# Characterizations and Kinetic Modelling of Boride Layers on Bohler K190 Steel

**Peter Orihel** [1], **Peter Jurči** [1,*] and **Mourad Keddam** [2]

1   Faculty of Material Sciences and Technology of the STU in Trnava, J. Bottu 25, 917 24 Trnava, Slovakia; peter.orihel@stuba.sk

2   Laboratoire de Technologie des Matériaux, Université des Sciences et de Technologie Houari Boumediène, Bab-Ezzouar 16111, Algeria; mkeddam@usthb.dz

\*   Correspondence: p.jurci@seznam.cz

**Abstract:** In this study, the Bohler K190 steel, manufactured by the powder metallurgy (PM) process, was subjected to the boronizing process. This thermochemical treatment was carried out in the range of 1173 to 1323 K, for 1–10 h. The scanning electron microscopy (SEM) was utilized for examining the morphology of layers' interfaces with a dual-phase nature and measuring the layers' thicknesses. The obtained boronized layers had a maximum thickness of $113 \pm 4.5$ μm. The X-ray diffraction analysis (XRD) confirmed the presence of FeB and $Fe_2B$ layers. The energy dispersive spectroscopy (EDS) mapping and EDS point analysis were used to investigate the redistribution of chemical elements within the boronized layers and the transition zone. The values of Vickers microhardness of $Fe_2B$, FeB, and transition zone were estimated. Finally, the boron activation energies in FeB and $Fe_2B$ were found to be 204.54 and 196.67 kJ·mol$^{-1}$ based on the integral method and compared to the literature results.

**Keywords:** boronizing; iron borides; kinetics; integral method; simulation

## 1. Introduction

Boronizing is a process of thermochemical treatment designed for enhancing the surface features of treated steels. It consists of diffusing the boron atoms via thermal process into the steel surfaces. The typical boronizing temperature is in the range of 1023–1323 K, for 0.5–10 h. When this process is applied, two iron borides FeB and $Fe_2B$ can be formed according to the controlling parameters, which are the time duration, the treatment temperature, and the quantity of boron source in the reactive medium. It is reported that the first iron boride to appear is the $Fe_2B$ phase [1], which possesses a tetragonal crystal lattice with boron content of 8.83 wt.% B. The typical values of hardness of this phase are approximately 1600 HV. The thermal expansion coefficient of this phase is $281 \times 10^{-6}$ K$^{-1}$. The FeB phase is formed as the second one and contains 16.23 wt.% B. The crystal lattice of this phase is orthorhombic and the typical value of hardness is approximately 2000 HV. The thermal expansion coefficient of FeB phase is $296 \times 10^{-6}$ K$^{-1}$. The remarkable properties of boride layers are high-surface hardness, resistance against wear, anti-corrosion resistance, low-friction coefficient, and low-fracture toughness.

The boronizing process is usually realized in gaseous [1,2] or liquid medium [3,4], in powder or paste [5,6], in plasma medium [7], or using electrolysis [8]. After this process, the boronized material is usually cooled down in the furnace. However, the re-austenitization, quenching, and tempering of boronized materials can be applied. The purpose of this heat treatment is to achieve the appropriate balance between toughness and strength [9].

In practice, boronizing in a powder mixture is the most frequently used method due to its simplicity and cost efficiency [10]. However, boronizing in plasma or by electrolysis is less time-consuming [11]. In the case of carbon steels, boride layers are formed according to the iron–boron equilibrium diagram [12]. Boride layers are usually biphased, and consist

of the FeB and $Fe_2B$ phases. However, especially in the case of carbon steels, boride layers can also be monophased, containing the sole $Fe_2B$ phase [11]. The thermal expansion coefficients of these phases are highly different, which can cause cracks formation at their interfaces. For this reason, the formation of monophased $Fe_2B$ layer is preferred in the industry [13]. The attainment of $Fe_2B$ layer, especially in the case of carbon steels, can be achieved by carefully controlling the boronizing parameters or by applying a diffusion annealing process after boronizing [14]. However, in the case of high-alloy steels, it is difficult to obtain the $Fe_2B$ layer solely. In this case, the thickness of FeB phase can reach 50% of the total layer thickness [15,16]. In the case of chromium steels, the chromium borides ($Cr_xB_y$) may be present in the boride layers as precipitates [17]. The thickness of boride layers is strongly dependent on the boronizing parameters. It increases as both the boronizing time and temperature rise. On the other hand, increasing carbon and alloying element contents limit the growth of boride layers by slowing down the mass flux of active boron. Moreover, this phenomenon is caused by the presence of metal borides (i.e., chromium borides) as precipitates that consume a part of the active boron during this thermodiffusion process.

For this reason, the morphology of boride layers formed on high-alloy steels is smoother and loses the typical saw-tooth morphology at interfaces with the substrate [18,19]. Some alloying elements have a stabilizing effect as silicon on ferrite. Therefore, in the case of steels with ferritic microstructure, the presence of silicon can create obstacles for the growth of boride layers, and deviations from the parabolic growth law of boronized layers can be observed [20].

Concerning the modelling of the boronizing process for iron-based alloys (i.e., steels), several approaches could be implemented to study the boron diffusion phenomenon. These modelling tools are of great significance in order to optimize the surface features of treated workpieces made of steels. Consequently, this optimization of surface properties could be achieved by appropriately selecting the layers' thicknesses that comply with the extreme working conditions. Through the existing literature, two microstructural configurations were considered for the iron boride coatings during the modelling of boron diffusion in the case of steels. Some reported models were implemented for the monophased layers of $Fe_2B$ type [21–26] or biphased layers (FeB + $Fe_2B$) [27–33]. One of the applied approaches is the phase-field method, which was already implemented by Ramdan et al. [24], in order to track the time evolution during the nucleation and growth of boride needles when generating compact $Fe_2B$ layers. Recently, Chen et al. [34] reviewed the potential applications of phase-field method for designing structural materials with appropriate properties. As a promising computational method, it allowed for the incorporation of different energetic contributions in terms of phase transformations into the phase-field equations to achieve this objective.

No kinetics studies have been devoted to the boronizing kinetics of PM Bohler K190 steel through the application of diffusion models. For this reason, the integral model [27,28] was implemented in this current study to assess the boron diffusion coefficients in FeB and $Fe_2B$ layers, and thereby deduce the boron activation energies in both phases (FeB and $Fe_2B$). The PM Bohler K190 steel is of primary importance from a practical point-of-view. It exhibits remarkable properties and possesses a very low proportion of inclusions when compared to the conventional tool steels. It is characterized by a homogeneous microstructure with fine grains providing the best combination between the property of toughness and resistance to compressive strength with an acceptable wear resistance. However, to withstand extreme wear conditions, this type of tool steel should be surface-hardened by boronizing to achieve a maximum wear resistance. To date, no study has been reported in the literature regarding the boronizing process of Bohler K190 steel to improve its surface features, especially its tribological behavior.

This paper aims to investigate the modification of surface features of pack-boronized Bohler K190 steel by employing the boronizing agent called Durborid in the interval of 1173–1323 K. The surface properties induced by boronizing were then investigated by

using adequate experimental tools. Particularly, the SEM observations were carried out to examine the morphology of generated interfaces along with the measurements of layers' thicknesses. The phase composition of boride layers was carried out with the help of XRD analysis. The property of surface hardness was examined by using the microhardness Vickers testing. Moreover, the redistribution of alloying elements between the boride layers and substrate was quantified by EDS mapping and point analysis. Furthermore, the integral diffusion model [27,28] was employed for obtaining the values of boron activation energies in FeB and $Fe_2B$ for PM Bohler K190 steel. This part of the kinetic study was completed by comparing our values of activation energies with literature data.

## 2. Materials and Methods

The used material was Bohler K190 steel, manufactured by the PM process (Bohler Edelstahl, Kapfenberg, Austria), with the chemical composition given in Table 1.

**Table 1.** Chemical composition of the Bohler K190 steel used in the current work.

| Element | C | Si | Mn | Cr | Mo | V |
|---|---|---|---|---|---|---|
| Content (wt.%) | 2.3 | 0.6 | 0.3 | 12.5 | 1.1 | 4.0 |

The samples were first grinded using SiC sandpapers with a grit of 600 and 1200 and polished with diamond paste of particle sizes of 6, 3, and 1 μm. After preparation, the samples were ultrasonically cleaned and degreased in acetone for 15 min. Before the boronizing process, the samples were placed into the steel container and covered with Durborid powder mixture (see Figure 1). Then, the samples with Durborid powder were hermetically sealed and the container was inserted into an electrical resistance furnace where it was heated to 1173, 1223, 1248, 1273, or 1323 K, and for 1, 3, 5, 7, or 10 h at each temperature. After the boronizing treatment, the container was removed from the furnace and the samples were cooled down to room temperature. Then, the specimens were cross-sectioned (with respect to the developed boride layers) and subjected to the standard metallographic preparation line. After the final step of polishing, the samples were etched in Nital etchant (3% solution of $HNO_3$ in ethyl alcohol) for 60 s. The microstructure of boride layers was analyzed by a scanning electron microscope (SEM) Jeol JSM-7600F (Tokyo, Japan). For analysis, the secondary electrons (SE) detection regime, at an acceleration voltage of 15 kV was used. Thicknesses of individual and total boride layers were estimated on randomly selected places. For a sufficient reliability of the results, the Kunst and Schaaber method has been used [35]. The basic principle of the measurement is depicted in Figure 2. The thickness values of FeB phase u as well as the thickness values of the entire FeB + $Fe_2B$ layer v were measured from the free surface. The mean values of the thicknesses were then calculated by using Equations (1) and (2).

$$u = \sum_{i=1}^{n} \frac{u_i}{n} \tag{1}$$

$$v = \sum_{i=1}^{n} \frac{v_i}{n} \tag{2}$$

For the quantification of elemental redistribution, the energy dispersive spectroscopy (EDS) was used. The mappings of chemical elements and point analysis (with a minimum of eight measurements in each boride compound) were realized. The XRD analysis of samples was obtained using a Phillips PW 1710 (Almelo, The Netherlands) with $Co_{K\alpha1,2}$ characteristic radiation, filtered by iron. The recording of diffracted intensities from XRD analysis was realized in the 2-theta angle between 10° and 100° with a step of 0.05°. However, the results from each X-ray pattern may represent the phase composition of the material surface and its substrate depending on the penetration depth of X-rays. The diffraction maxima were identified using the HighScore Plus program version 3.0.5. The microhardness values of boride layers, diffusion zone, and substrate were obtained by

using a Hanneman microhardness tester (Jena, Germany), with a load of 100 g ($HV_{0.1}$) during a loading time of 15 h. To obtain the relevant information on Vickers microhardness, seven measurements were made in each place.

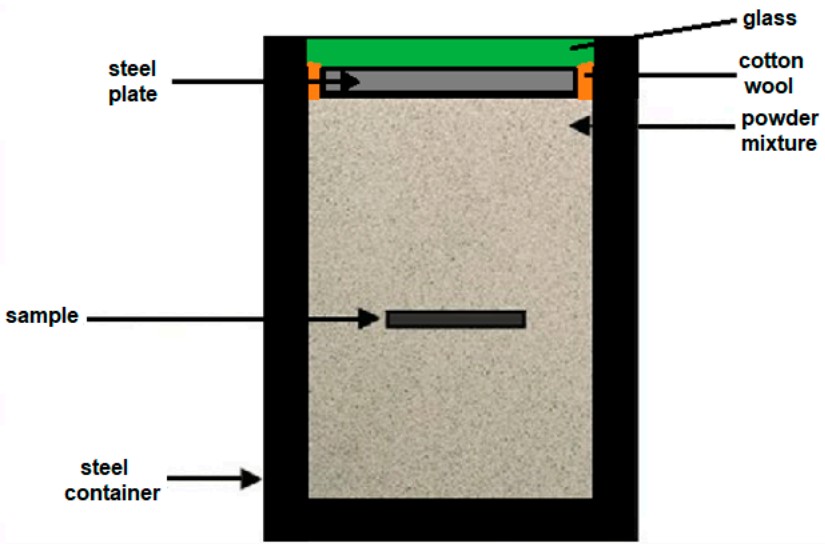

**Figure 1.** Schematization of the container with its components designed for the powder-pack boronizing process.

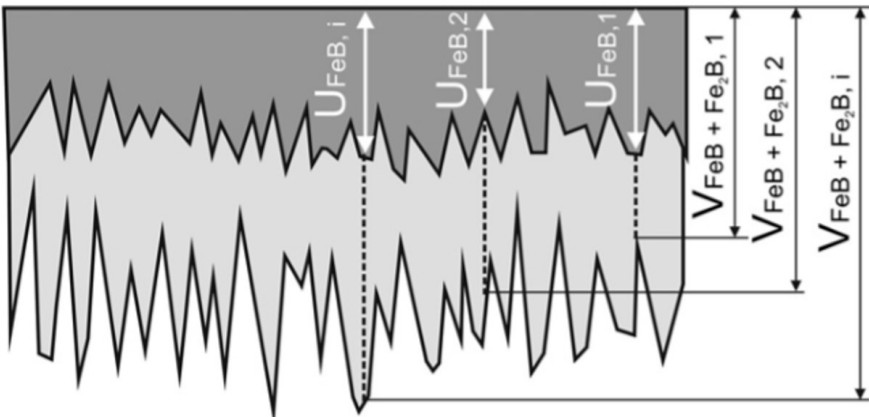

**Figure 2.** A schematic view of cross-sectioned specimens showing the procedure of the measurements of boronized layer thickness.

### 3. The Integral Diffusion Model

The integral diffusion model was implemented in this current study to analyze the boronizing kinetics of Bohler K190 steel. The same approach was already applied for modelling the boronizing kinetics of Royalloy and X165CrV12 steels [27,28]. It was used to simulate the time dependencies of layers' thicknesses of FeB and (FeB + $Fe_2B$) after assessing the values of boron diffusion coefficients in the FeB and $Fe_2B$ phases. The diffusion of boron atoms occurs in the semi-infinite medium within the steel matrix saturated with boron atoms. Figure 3 gives a schematic view of the generated boron concentration profiles across the FeB and $Fe_2B$ layers without the occurrence of boride incubation periods.

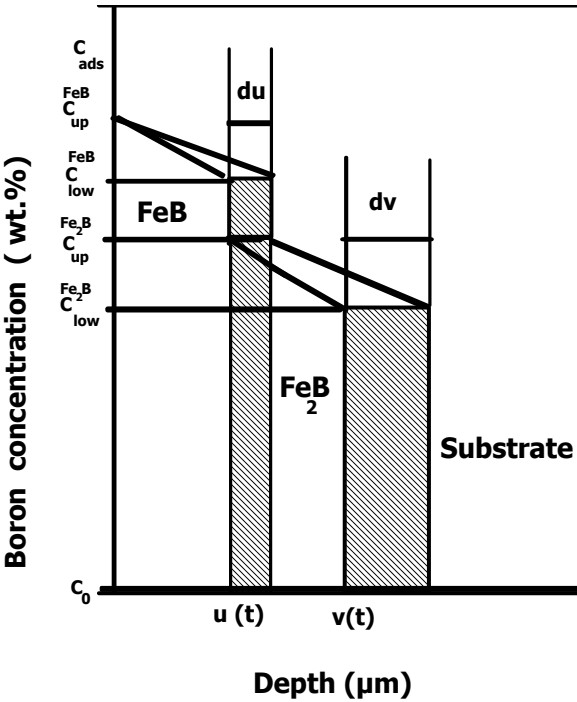

**Figure 3.** Schematization of the boron concentration profiles in the case of the bilayer (FeB + Fe$_2$B).

The equilibrium boron concentrations with upper and lower limits in FeB and Fe$_2$B are represented in Figure 1 with $C_{up}^{FeB}$ (=16.40 wt.% B) and $C_{low}^{FeB}$ (=16.23 wt.% B) for the FeB phase. For the Fe$_2$B phase, the corresponding values are $C_{up}^{Fe_2B}$ (=9 wt.% B) and $C_{low}^{Fe_2B}$ (=8.83 wt.% B) [27,28]. The adsorbed amount of boron at the steel surface is designated by the quantity $C_{ads}$ [30]. The variable $x = u(t)$ designates the position of the first (FeB/Fe$_2$B) interface, while $x = v(t)$ refers to the second (Fe$_2$B/substrate) interface. The solubility limit of boron atoms within the steel matrix is represented by the concentration $C_0$ and equal to $35 \times 10^{-4}$ wt.% B [12,36]. Equation (3) represents the change in time of the FeB layer thickness $u(t)$:

$$u(t) = k'\sqrt{t} = 2\varepsilon\sqrt{D_{FeB}t} \tag{3}$$

where $k'$ denotes the kinetic constant at the (FeB/Fe$_2$B) interface and $\varepsilon$ is the associated dimensionless parameter related to the boron diffusion coefficient in FeB. Equation (4) gives the time dependencies of the entire layer (FeB + Fe$_2$B):

$$v(t) = k\sqrt{t} = 2\eta\sqrt{D_{Fe_2B}t} \tag{4}$$

where $k$ denotes the parabolic growth constant at the (FeB/Fe$_2$B) interface and $\eta$ is the second dimensionless parameter related to the boron diffusion coefficient in Fe$_2$B. The assumptions considered while establishing the mathematical foundation of this kinetic approach are the following: (i) The diffusion of boron atoms is a one-dimensional problem, (ii) the layer thickness is small in comparison with the sample dimension, (iii) the boron concentrations at interfaces are independent of time, (iv) the boron diffusion coefficient in each phase obeys the Arrhenius relationship, and (v) the process temperature remains constant during the treatment. In the integral method, the boron concentration profiles in the FeB and Fe$_2$B layers are expressed by Equations (5) and (6) following the method proposed by Goodman [37]:

$$C_{FeB}(x,t) = C_{low}^{FeB} + a_1(t)(u(t) - x) + b_1(t)(u(t) - x)^2 \text{ for } 0 \leq x \leq u \tag{5}$$

$$C_{Fe_2B}(x,t) = C_{low}^{Fe_2B} + a_2(t)(v(t) - x) + b_2(t)(v(t) - x)^2 \text{ for } u \leq x \leq v \tag{6}$$

The time-dependent parameters $a_1(t)$, $a_2(t)$, $b_1(t)$, and $b_2(t)$ must verify the boundary conditions. Therefore, the integral method is based on the numerical solving of the set of differential algebraic equations (DAE) given by Equations (7)–(12).

$$a_1(t)u(t) + b_1(t)u(t)^2 = (C_{up}^{FeB} - C_{low}^{FeB}) \tag{7}$$

$$a_2(t)(v(t) - u(t)) + b_2(t)(v(t) - u(t))^2 = (C_{up}^{Fe_2B} - C_{low}^{Fe_2B}) \tag{8}$$

$$\frac{d}{dt}\left[\frac{u(t)^2}{2}a_1(t) + \frac{u(t)^3}{3}b_1(t)\right] = 2D_{FeB}b_1(t)u(t) \tag{9}$$

$$\begin{aligned}2w_{12}\frac{dv(t)}{dt} + \frac{(v(t)-u(t))^2}{2}\frac{da_2(t)}{dt} + \\ \frac{(v(t)-u(t))^3}{3}\frac{db_2(t)}{dt} = 2D_{Fe_2B}b_2(t)(v(t) - u(t))\end{aligned} \tag{10}$$

$$[a_1^2(t) - 2w_1b_1(t)]D_{FeB} = a_1(t)[a_2(t) + 2b_2(t)(v(t) - u(t))]D_{Fe_2B} \tag{11}$$

$$2w_{12}a_2(t)b_1(t)D_{FeB} = a_1(t)[a_2^2(t) - 2w_2b_2(t)]D_{Fe_2B} \tag{12}$$

with $w_1 = \left[\frac{(C_{up}^{FeB}+C_{low}^{FeB})}{2} - C_{up}^{Fe_2B})\right]$, $w_2 = \left[\frac{(C_{up}^{Fe_2B}+C_{low}^{Fe_2B})}{2} - C_0)\right]$ and $w_{12} = \frac{(C_{up}^{Fe_2B}-C_{low}^{Fe_2B})}{2}$.

By choosing appropriate changes in variables [27,28], the DAE system represented by Equations (7)–(12) can be turned into a set of non-linear equations to find the numerical values of $\alpha_1$, $\beta_1$, $\alpha_2$, and $\beta_2$ constants by using the Newton–Raphson method [38], which is needed for the estimation of boron diffusion coefficients in FeB and Fe$_2$B. Consequently, the values of dimensionless parameters $\varepsilon$ and $\eta$ can be readily obtained by using Equations (13) and (14):

$$\varepsilon = \sqrt{\frac{\beta_1}{\left(\frac{\alpha_1}{2} + \frac{\beta_1}{3}\right)}} \tag{13}$$

and

$$\eta = k\sqrt{\frac{\beta_2}{[2w_{12}k(k - k') - (\frac{\alpha_2}{2} + \frac{2\beta_2}{3})(k - k')^2]}} \tag{14}$$

The values of boron diffusion coefficients in FeB and Fe$_2$B are deduced from Equations (15) and (16):

$$D_{FeB} = \left(\frac{k'}{2\varepsilon}\right)^2 \tag{15}$$

and

$$D_{Fe_2B} = \left(\frac{k}{2\eta}\right)^2 \tag{16}$$

## 4. Results and Discussion

### 4.1. SEM Examinations and EDS Analysis

The cross-sectional SEM images of differently borided specimens are shown in Figure 4. It is clearly visible that all boride layers are biphased and consist of the FeB and Fe$_2$B phases. However, in the case of 1173 K for 1 h, the FeB layer does not appear and only the Fe$_2$B layer is formed. The interfaces have a high tendency to flatness as observed in borided Royalloy steel [27]. This scenario is ascribed to the effect of alloying elements present in the steel matrix with the production of dense and thicker layers for a prolonged time duration (10 h).

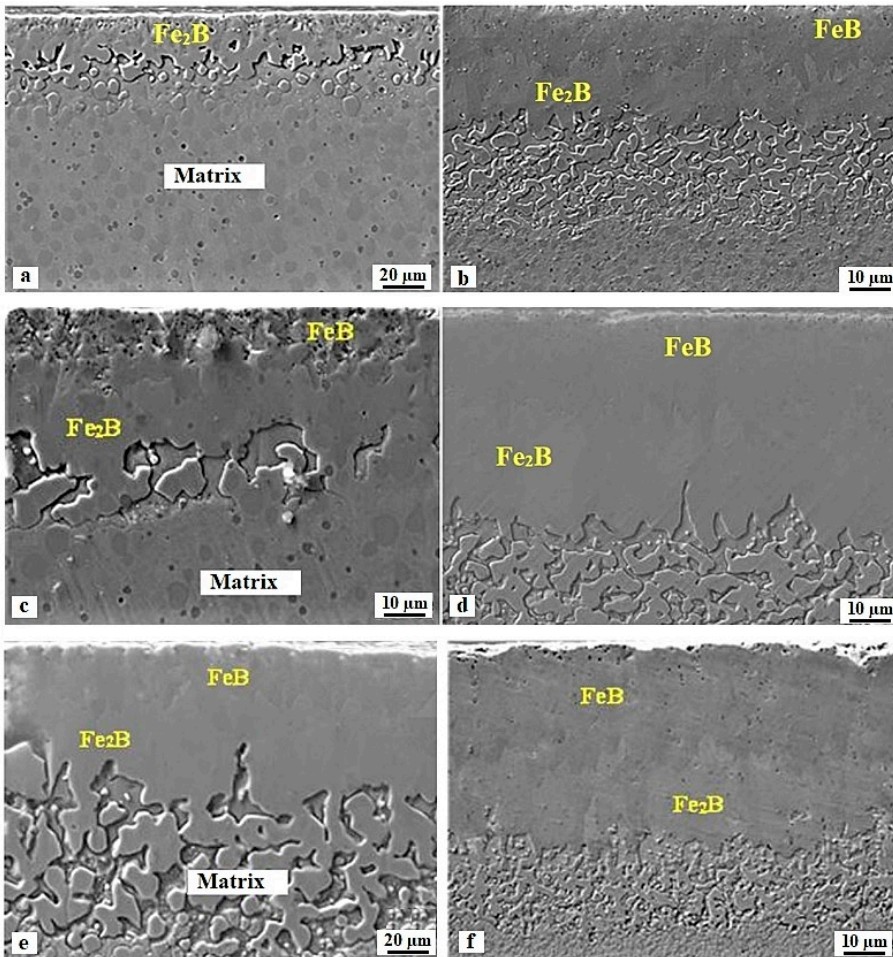

**Figure 4.** Cross-sectional SEM images of samples boronized at different processing conditions: (**a**) 1173 K for 1 h, (**b**) 1173 K for 10 h, (**c**) 1248 K for 1 h, (**d**) 1248 K for 10 h, (**e**) 1323 K for 1 h, and (**f**) 1323 K for 10 h.

Figure 5 shows the results of EDS mapping of boronized (at 1223 K for 10 h) Bohler K190 steel. In this case, it is worth noting that similar elemental redistribution takes place in all the boronized specimens, and that the specimen differs from one to another only by the extent of this redistribution. A strong redistribution of alloying elements was noticed during the boronizing process. The chromium is the most redistributed element from the substrate underneath the boride layers, to the transient region, which is accompanied with the formation of additional particles in the close-to-boride region. These particles contained the highest content of chromium. Additionally, the silicon is insoluble in the borides, and hinders the diffusion of boron atoms toward the steel substrate. Therefore, the silicon element is being accumulated at the $Fe_2B$/substrate interface and reached a maximum concentration of 5.07 wt.%. Moreover, this experimental outcome was visible in reference [39], in the case of boronized ASTM A36 steel with the following chemical compositions (in wt.%) 0.1% C, 0.20% Si, 0.85% Mn, 0.20% Cu, 0.040% P, and 0.050% S. In this paper, we demonstrate that silicon was expelled from the surface layer, as a result of borides' growth, to the nearby $Fe_2B$/substrate interface forming the Fe-Si-B compounds ($FeSi_{0.4}B_{0.6}$ and $Fe_5SiB_2$) [39].

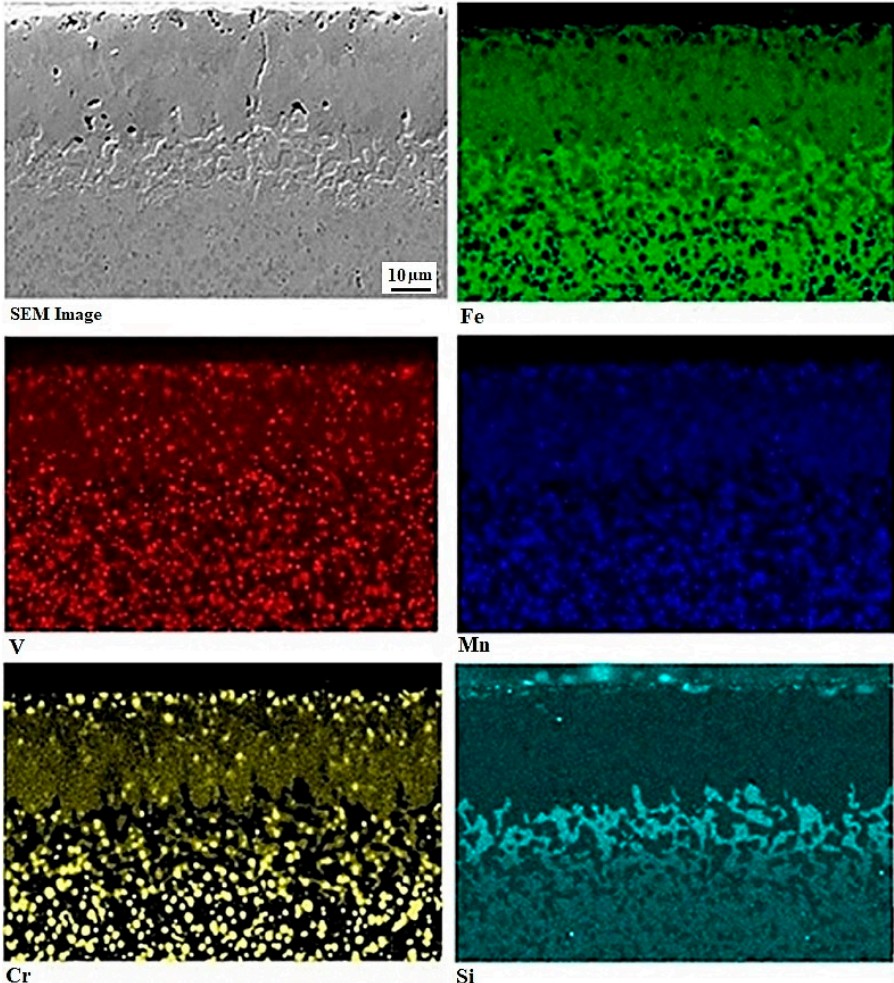

**Figure 5.** Cross-sectional view of boronized sample at 1223 K for 10 h along with the EDS mapping of Fe, Cr, V, Mn, and Si elements.

To quantify the redistribution of alloying elements during the boronizing process, the EDS point analysis was realized. The results of this analysis are shown in Figure 6. It It is clearly visible that the chromium content in the boride layers is slightly higher in comparison with the substrate. This is proof that the chromium is displaced from the substrate to the boride layers by leaving the region near the boride layers. This region is at the same time depleted of it. The $Fe_2B$ phase contains a slightly higher chromium content in comparison with the FeB phase. The reason for this phenomenon is probably the gradual diffusion of chromium from the substrate to the boride layers or its easy incorporation into the $Fe_2B$ phase in comparison with the FeB phase. It is known that the $Fe_2B$ phase is first formed on the surface during boronizing [40]. Therefore, the chromium diffuses first into the $Fe_2B$ and only after a certain period into the FeB. In a study by Fellner and Chrenkova [41], different carbon- and high-chromium ledeburitic steels were borided in a molten mixture of boron carbide and borax, and achieved very similar outcomes and explanations. In a study by Dybkov [16], steels were borided with different Cr contents and reported a significantly higher chromium content in $Fe_2B$ than in FeB. In addition to the fact that the $Fe_2B$ is first enriched by the chromium diffusion from the substrate, the isomorphism of the two phases $Fe_2B$ and $Cr_2B$ should be considered [41]. Based on this fact, a partial substitution of Fe atoms by Cr atoms is rendered possible when forming the assumed ternary boride. Conversely, the borides FeB and CrB are not isomorphous and, in this case, the formation of a ternary boride is unlikely. This may also contribute to explain the lower chromium content observed in the FeB phase in comparison with the $Fe_2B$ phase.

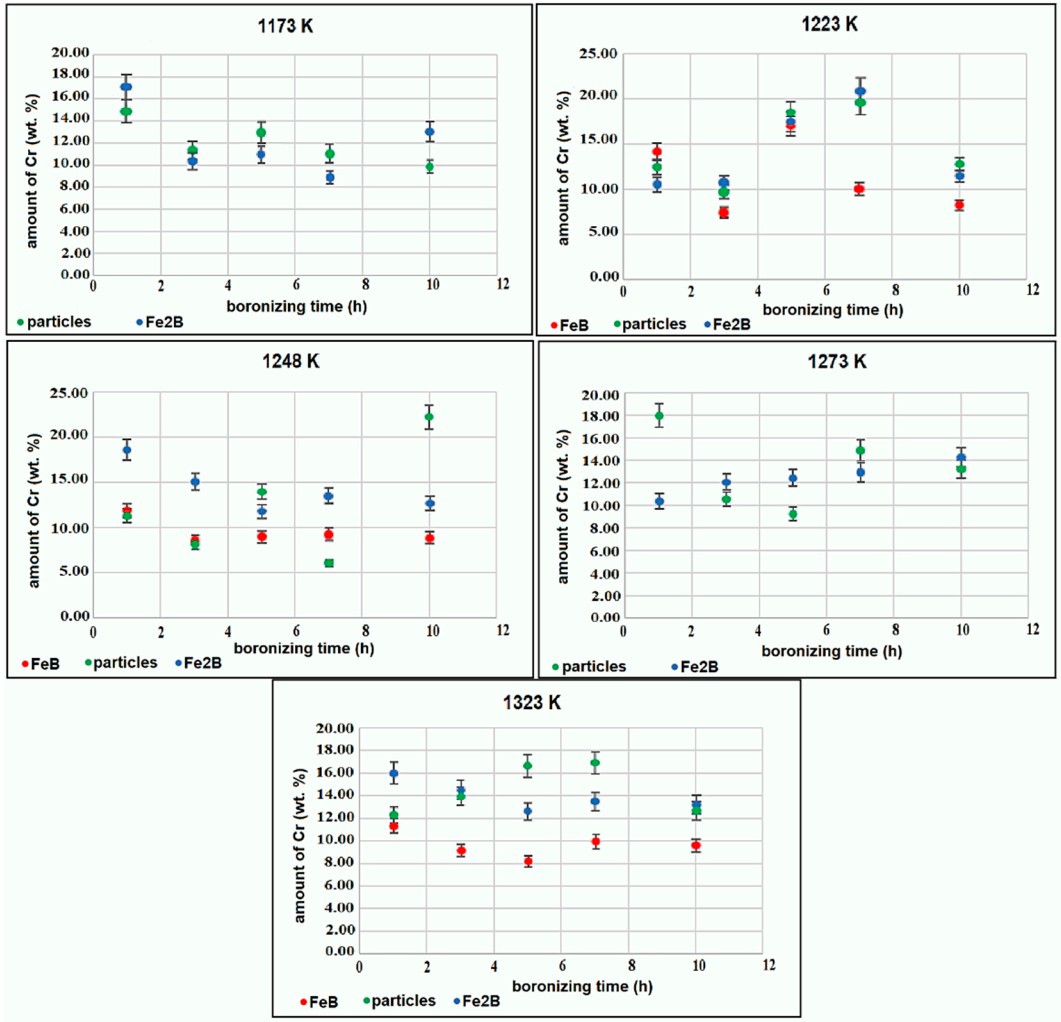

**Figure 6.** Chromium contents in FeB, Fe$_2$B, and in particles underneath the boride layers.

In all cases, the particles formed in the transient region contain the highest content of chromium, approximately 20–25 wt.%. The highest amount of silicon was obtained in the transient region underneath the boride layers, as shown in Figure 7. The maximum value of silicon content ranged between 3.73 wt.% and 5.07 wt.%. The boronizing temperature has a limited effect only on the maximum content of silicon within the transition zone. This result is in good agreement with the experimental findings obtained on the boronized Royalloy steel [27].

*4.2. XRD Results*

To support the results of microscopic examinations of cross-sectional views of boronized Bohler K190 steel, the experimental characterization by XRD technique is crucial to identify the phases formed at the surface layer. The XRD patterns of boronized samples at temperatures of 1173, 1248, and 1323 K for either 1 or 10 h, are shown in Figure 8. It is clearly visible that all the boride layers are biphased and consist of the FeB and Fe$_2$B phases. However, borides of alloying elements were not detected. It is known that in the case of high-alloy steels, the metal borides, such as chromium borides can be present in the boride layers. Chromium borides were obtained, for instance, in the case of AISI 440C steel (2.1 wt.% C; 16.50 wt.% Cr; 0.417 wt.% Mn) [10], boronized in Ekabor II powder mixture. The chromium borides CrB and Cr$_2$B were also obtained within the boride layers on the AISI D2 steel with the following chemical compositions: 0.90 wt.% C; 7.80 wt.% Cr; and 2.50 wt.% Mo [42]. Moreover, the results of XRD analysis for boronized AISI D2 steel

revealed the formation of molybdenum borides (MoB and $Mo_4B_2$), vanadium boride ($V_2B$) as precipitates, in addition to the $Fe_2B$ phase [42]. In reference [13], the CrB phase was identified by XRD analysis in the boride layers for Vanadis 6 steel while employing other processing parameters (950, 1000, and 1000 °C for 45 min and 10 h). However, in the case of boronized Royalloy steel [27], the chromium borides were not detected in the boride layers by XRD analysis. This fact was attributed to the overlapping of diffraction peaks of iron borides with those of chromium borides $Cr_xB_y$, which makes their deconvolution rather difficult. To accurately detect the presence of chromium borides inside the boride layers, the use of transmission electron microscopy is strongly recommended. Therefore, it is necessary to prepare thin foils of boride layers in order to identify their nature and chemical composition.

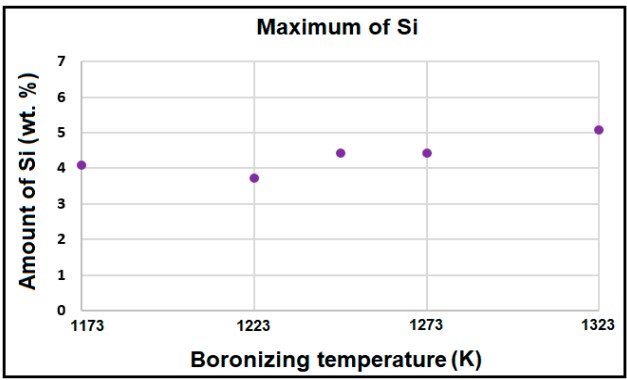

**Figure 7.** Maximum content of silicon in the transient region underneath the boride layers.

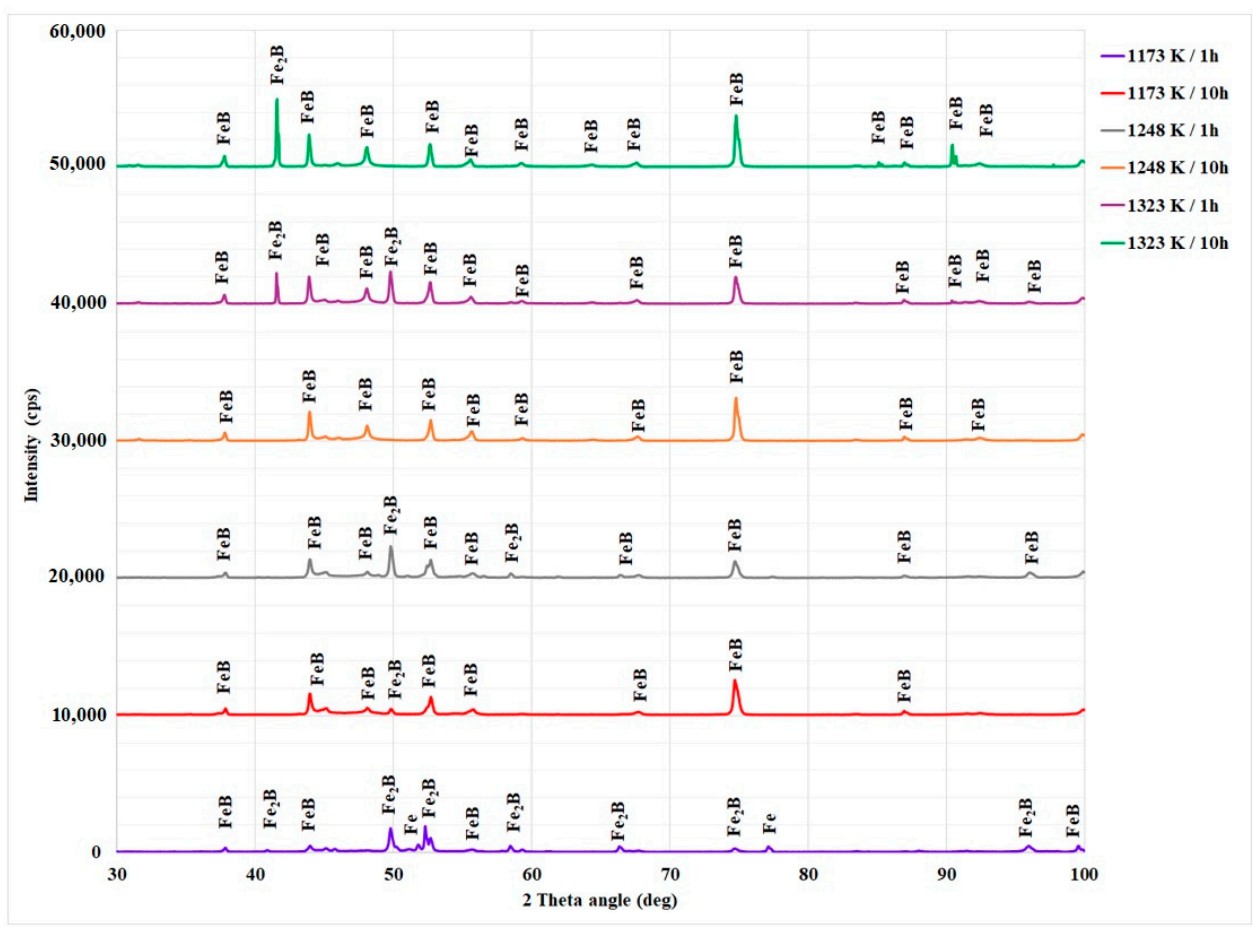

**Figure 8.** X-ray diffraction patterns of boride layers on Bohler K190 steel.

### 4.3. Vickers Microhardness Measurements

　　The microhardness of boron compounds is very crucial for their wear performance. A success of the thermochemical treatment, such as boronizing, depends on the degree of hardening attained in the surface layer to generate hard phases by thermodiffusion. Therefore, the establishment of microhardness profiles exhibiting a gradient of this property along the depth is a key factor for determining the efficiency of boronizing process. The Vickers microhardness $HV_{0.1}$ of boride layers and transient region are shown in Figure 9. The higher microhardness value was recorded for the FeB layer on the outer surface of boronized samples. The values of microhardness for the FeB layer ranged between 1992 and 2245 $HV_{0.1}$. The microhardness of the $Fe_2B$ phase was lower than the FeB phase and ranged between 1579 and 1743 $HV_{0.1}$. The Vickers microhardness of transient region was in the range of 748–1141 $HV_{0.1}$. Moreover, it is clearly visible that the microhardness depends only slightly on the temperature of boronizing or its duration, and the magnitudes are very similar to those obtained on Royalloy [27] or Vanadis 6 steels [13].

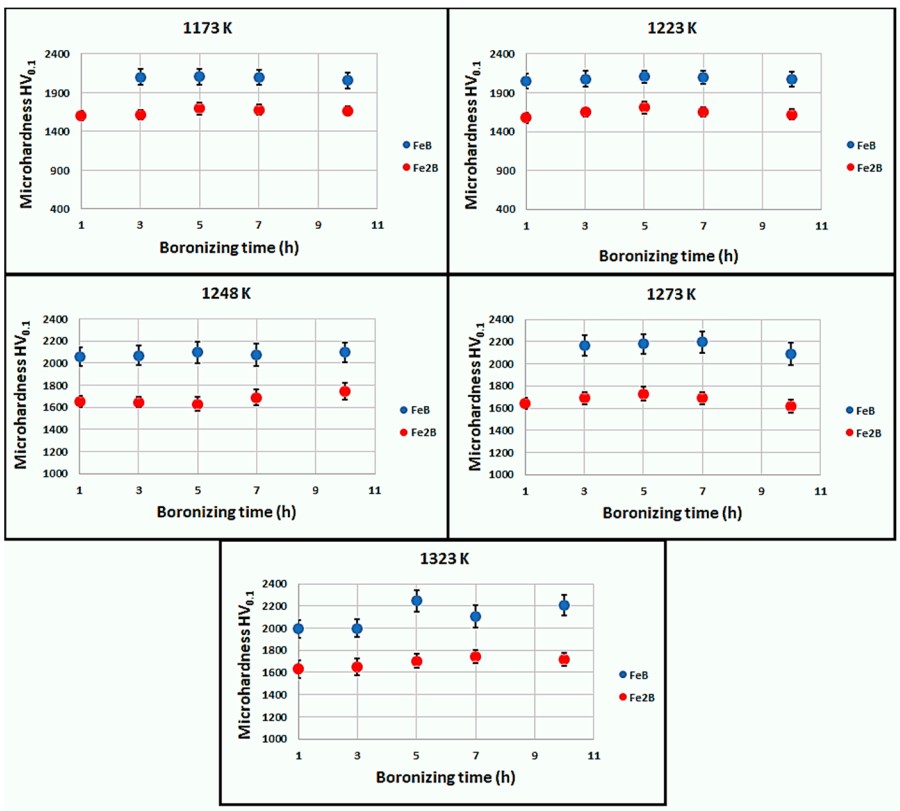

**Figure 9.** Measured values of Vickers microhardness in the three locations (FeB, $Fe_2B$, and diffusion zone) on the cross-sections of boronized layers for different conditions.

### 4.4. Assessment of Boron Diffusion Coefficients in Iron Borides with the Integral Method

　　The determination of boron diffusivity in each iron boride (FeB or $Fe_2B$) is a crucial step, which allows for the modelling of boronizing kinetics of Bohler K190 steel for the selected processing parameters. The employed model assumes the diffusion of active boron at the atomic level to develop a biphased boride layer that consists of FeB and $Fe_2B$, in which the boron concentrations remain constant at the dual-phase interfaces and independent of time duration. Therefore, the plots of time dependencies of layers' thicknesses permit the extraction of experimental parabolic constants needed for assessing the boron diffusivities in iron borides. Figure 10 describes the change in time duration of experimental layers' thicknesses obtained on the Bohler K190 steel. It is clear from the plotted straight lines of Figure 10 that the layers generated by boronizing obeyed the classical parabolic law, and the process is then controlled by the diffusion of boron atoms at the atomic level. In

Table 2, the experimental values of parabolic growth constants, obtained from the slopes of corresponding straight lines in the temperature range of 1173–1323 K, are grouped.

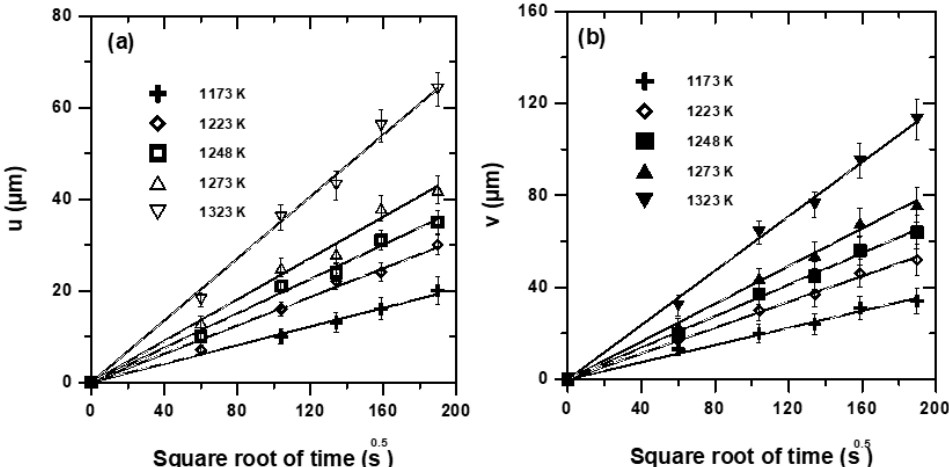

**Figure 10.** Time dependencies of layers' thicknesses: (**a**) The FeB layer, (**b**) the bilayer (FeB + Fe$_2$B).

**Table 2.** Experimental kinetics constants determined at the phase interfaces in the interval of 1173 to 1323 K.

| Temperature (K) | $k'$ ($\mu$m·s$^{-0.5}$) at the First Phase Interface | $k$ ($\mu$m·s$^{-0.5}$) at the Second Phase Interface |
|---|---|---|
| 1173 | 0.1013 | 0.1864 |
| 1223 | 0.1559 | 0.2806 |
| 1248 | 0.1876 | 0.3431 |
| 1273 | 0.2258 | 0.4103 |
| 1323 | 0.3378 | 0.5906 |

The set of non-linear equations stemming from the DAE system was solved numerically via the Newton–Raphson routine [38] to obtain the following values $\alpha_1$, $\beta_1$, $\alpha_2$, and $\beta_2$ related to the two dimensionless parameters $\varepsilon$ and $\eta$. Table 3 summarizes the calculation results regarding the values of boron diffusion coefficients in both phases by employing Equations (15) and (16) and the values of the two dimensionless parameters $\varepsilon$ and $\eta$. From Table 3, it is visible that these two parameters are not affected by the processing temperature and remained constant within the considered temperature range, which demonstrated the parabolic nature of iron boride layers' growth.

**Table 3.** Calculated boron diffusion coefficients in iron borides using the integral diffusion model.

| T (K) | $D_{FeB}$ ($\times 10^{-12}$ m$^2$·s$^{-1}$) Equation (15) | $D_{Fe_2B}$ ($\times 10^{-12}$ m$^2$·s$^{-1}$) Equation (16) | $\varepsilon$ Parameter | $\eta$ Parameter |
|---|---|---|---|---|
| 1173 | 0.50 | 0.32 | 0.0712 | 0.1628 |
| 1223 | 1.18 | 0.73 | 0.0716 | 0.1643 |
| 1248 | 1.73 | 1.10 | 0.0713 | 0.1632 |
| 1273 | 2.49 | 1.57 | 0.0714 | 0.1636 |
| 1323 | 5.48 | 3.14 | 0.0721 | 0.1666 |

The calculated results in terms of boron diffusivities in iron borides displayed in Table 3 were fitted by employing the Arrhenius relations to obtain, from the slopes of plotted lines, the values of boron activation energies in both phases. The results of this fitting are shown in Equations (17) and (18):

$$D_{FeB} = 6.36 \times 10^{-4} \exp\left(\frac{-204.54 \text{ kJ/mol}}{RT}\right) \tag{17}$$

$$D_{Fe_2B} = 1.84 \times 10^{-4} \exp\left(\frac{-196.67 \text{ kJ/mol}}{RT}\right) \tag{18}$$

where $R$ represents the universal gas constant (8.314 J mol$^{-1}$·K$^{-1}$) and $T$ is the processing temperature given in Kelvin.

The values of boron activation energies estimated from the present work were compared to other data reported in the literature in the case of boronized high-alloy steels [27,28,43–48], as displayed in Table 4. It is clearly visible that the boron activation energies calculated or measured on these various materials are influenced by the key factors which include the following: The difference in chemical composition of substrate, the boriding process used to generate the boride coatings, the processing parameters, the nature of boron source, the calculation method or approach, and the nature of chemical reactions controlling the process. The high values of boron activation energies in FeB and Fe$_2$B are attributed to the effect of alloying elements present in the Bohler K190 steel. They are consistent with the literature data in the case of pack-boronizing process [27,28,43–48]. In fact, the increase in alloying elements in the matrix steel tend to hinder the boron diffusion and act as a diffusion barrier that results in a reduction in layers' thicknesses. For example, in a study by Makuch et al. [28], the X165CrV12 was pack-boronized with the powder mixture comprised of 50 wt.% B$_4$C, 0.5 wt.% AlF$_3$, and 49.5 wt.% Al$_2$O$_3$ to generate the FeB and Fe$_2$B layers. The estimated boron activation energy in FeB was lower than the Fe$_2$B phase, in contrast to other literature results [27,43,47,49], in which the Fe$_2$B phase was formed before the FeB phase. However, Makuch et al. [28] claimed that the FeB phase was formed first via phase transformation, similar to the phase observed in the gas boriding process [50]. In a study by Keddam et al. [51], the AISI 440C steel was plasma-paste boronized by employing the borax paste as a boron source. From Table 4, the obtained activation energy was the lowest (=134.62 kJ·mol$^{-1}$) in comparison with those obtained or estimated in the case of powder-pack boriding [27,28,43–48]. This situation is certainly ascribed to the activation of different species inside the generated plasma. In another study, Kartal et al. [52] designed a recent cost-efficient method of surface hardening referred to as CRTD-Bor. This process was used to form solely the Fe$_2$B phase by combining the action of CRTD with the phase homogenization (PH) step. The employed electrolyte contained 90 wt.% Na$_2$B$_4$O$_7$ and 10 wt.% Na$_2$CO$_3$. XRD studies indicated the presence of only FeB for processing temperatures of 850, 900, and 950 °C during 15 min, whereas for higher temperatures (i.e., 1000 and 1050 °C), both iron borides co-exist. The boron activation energy was then estimated by using the classical parabolic growth law independently of the phase composition of boronized layers and the authors obtained a value of 179 kJ·mol$^{-1}$. This latter value of activation energy is lower than those obtained in the case of pack-boronizing of AISI T1 steel [50] (=212.76 kJ·mol$^{-1}$), since in the CRTD-Bor, the electrochemical reactions proceed rapidly, and thus reduce the activation energy of the system. Campos-Silva et al. [53] designed a new boronizing method referred to as the pulsed-DC powder-pack boriding process (PDCPB). This surface-hardening technique is promising due to shortening the time duration and saving energy. It was applied to the AISI 316L steel substrates to form the bilayer (FeB/Fe$_2$B) under a constant current input of 5 A with the possibility of changing the polarity in the cathodes. This process led to the reduction in boron activation energies in FeB and Fe$_2$B in comparison with the conventional powder methods [27,28,43–48].

In another work, Campos et al. [49] employed the paste-boronizing treatment to surface-harden the AISI M2 steel. The result of this thermochemical process led to the formation of FeB, Fe$_2$B, and diffusion zone in the interval of 1123 to 1273 K. The effect of boron-paste thickness on the kinetics was evidenced by varying its value from 3 to 5 mm. An increase in boron-paste thickness resulted in the rise of the parabolic growth constants at the three growing phase interfaces. In addition, the bilayer model was suggested to determine the respective boron activation energies in FeB and Fe$_2$B, which were 257.5 and 201 kJ·mol$^{-1}$.

**Table 4.** Comparison of the estimated boron activation energies with those from the literature for borided high-alloy steels.

| Steel | Boriding Process | Operating Parameters | Phases Present | Activation Energy (kJ·mol$^{-1}$) | Calculation Method | Refs. |
|---|---|---|---|---|---|---|
| AISI 440 C | PPB | 700–800 °C For 3–7 h | FeB, Fe$_2$B, CrB, Cr$_2$B | 134.62 | Parabolic growth law | [51] |
| AISI TI | CRTD-Bor | 850–1050 °C For 0.25–1 h | FeB and/or Fe$_2$B | 179.05 | Parabolic growth law | [52] |
| AISI 316 L | PDCPB | 850–950 °C For 0.5–2 h | FeB, Fe$_2$B, CrB, Cr$_2$B | 162.7 ± 7 (FeB) 171 ± 5 (Fe$_2$B) | Bilayer model | [53] |
| AISI M2 | Paste | 950–1000 °C for 2 and 6 h | FeB, Fe$_2$B | 257.5 (FeB) 201 (Fe$_2$B) | Bilayer model | [51] |
| SS410 | Powder | 850–1000 °C For 2–8 h | No reported | 186.49 | Parabolic growth law | [44] |
| AISI D2 | Powder | 850–1000 °C For 2–8 h | Fe$_2$B | 201.5 | Monolayer model | [45] |
| AISI 304 | Powder | 850–1050 °C For 1–4 h | FeB, Fe$_2$B, Ni$_2$B, Cr$_2$Ni$_3$B$_6$ | 244 | Parabolic law | [46] |
| AISI H13 | Powder | 800–1000 °C For 2–6 h | FeB, Fe$_2$B, CrB, Cr$_2$B | 236.43 (FeB) 233.04 (Fe$_2$B) | MDC method | [47] |
| ASP®2012 | Powder | 850–950 °C For 2–6 h | FeB, Fe$_2$B, CrB, Mo$_2$B, and W$_2$B | 314.716 | Parabolic growth law | [48] |
| Royalloy | Powder | 900–1050 °C for 1–10 h | FeB, Fe$_2$B | 242.79 (FeB) 223.0 (Fe$_2$B) | Integral method | [27] |
| X165CrV12 | Powder | 850–950 °C For 3–9 h | FeB, Fe$_2$B, CrB | 173.73 (FeB) 193.47 (Fe$_2$B) | Integral method | [28] |
| AISI M2 | Powder | 850–950 °C for 2–6 h and 10 h | FeB, Fe$_2$B, CrB, Cr$_2$B, B$_4$V$_3$ | 206.41 (FeB) 216.18 (Fe$_2$B) | Integral method | [43] |
| AISI M2 | Powder | 850–950 °C for 2–6 h and 10 h | FeB, Fe$_2$B, CrB, Cr$_2$B, B$_4$V$_3$ | 226.02 (FeB) 209.04 (Fe$_2$B) | Dybkov model | [43] |
| Bohler K190 | Powder | 900–1050 °C for 1–10 h | FeB, Fe$_2$B | 204.54 (FeB) 196.67 (Fe$_2$B) | Integral method | This study |

In a study by Ramakrishnan et al. [44], the martensitic stainless steel (grade SS410) was pack-boronized at temperatures between 900 and 950 °C. The change in diameter of cylindrical specimens before and after boronizing was measured in the range of 0.12 to 0.36%, resulting in the increase in surface roughness of treated specimens. In addition, the boron activation energy in SS410 steel was calculated with the use of an empirical relationship and the authors obtained a value of 186.49 kJ·mol$^{-1}$. Nait Abdellah et al. [47] used the powder mixture of 90 wt.% B$_4$C and 10 wt.% NaBF$_4$ to produce the bilayer (FeB/Fe$_2$B) on AISI H13 steel in the range of 800–1000 °C. The mean diffusion coefficient (MDC) method was established to extract the boron activation energies in FeB and Fe$_2$B, which were 236.43 and 233.04 kJ·mol$^{-1}$, respectively. Additionally, the same model was verified empirically for two additional processing parameters (925 °C for 1 and 3 h).

As a limiting factor, the present approach did not account for the precipitation of metal borides. Apart from the formation of iron borides as compact layers, a certain content of boron atoms reacts with transition metals to give rise to the metal borides present as precipitates within the boronized layer. This situation leads to slowing down the diffusion rate of boron atoms, and thereby diminishing the layers' thicknesses. The carbon–boron reaction is neglected. In reality, the boron element competes with the carbon present in the

steel substrate to occupy the majority of octahedral sites in the iron lattice. As a result, the carbon element is pushed away from the diffusion front to pass into the diffusion zone [54]. Even though these two limitations are important, the integral method is a key tool that can be used to model the boronizing kinetics of any alloyed steel.

## 5. Conclusions

The Bohler K190 steel was treated thermochemically by employing the Durborid powder mixture in the interval of 1173 to 1323 K. The entire boride layer reached a maximum thickness of $113 \pm 4.5$ μm. The borides/matrix interfaces are relatively flat and smooth, which is typical for high-carbon and high-alloy steels. The XRD studies corroborated the presence of two iron borides (FeB and $Fe_2B$) for almost all (with the exception of 1173 K for 1 h) processing parameters. The repartition of alloying elements was put into practice through the EDS mapping. Chromium was found to be more soluble in $Fe_2B$ than in FeB. The particles that appeared in the transition zone were found to be rich in chromium content of approximately 20–25 wt.%. Silicon was more accumulated underneath the borides where its maximum content was in the range of 3.73–5.07 wt.%. The values of Vickers microhardness of FeB phase were located between 1992 and 2245 $HV_{0.1}$, while those of $Fe_2B$ were in the range of 1579–1743 $HV_{0.1}$. The Vickers microhardness values measured in the transition zone ranged from 748 to 1141 $HV_{0.1}$. The obtained values of microhardness were only slightly influenced by the processing parameters. Finally, the obtained values of boron activation energies in FeB and $Fe_2B$ were 204.54 and 196.67 kJ·mol$^{-1}$, respectively. These two activation energies were obtained from the integral method and deemed to be concordant with the literature results. For future works, the integral model can be extended to the multiphase system in which interstitial elements, such as boron, carbon, or nitrogen can generate hard and compact layers by analyzing the kinetics of their formation.

**Author Contributions:** Conceptualization, P.J. and M.K.; methodology, P.J. and M.K.; software, M.K.; validation, M.K. and P.J.; formal analysis, P.O.; investigation, P.O.; resources, not applicable; data curation, P.O. and M.K.; writing—original draft preparation, P.O. and M.K.; writing—review and editing, P.J. and M.K.; visualization, P.J., M.K. and P.O.; supervision, P.J.; not applicable; funding acquisition, not applicable. All authors have read and agreed to the published version of the manuscript.

**Funding:** This research received no external funding.

**Institutional Review Board Statement:** Not applicable.

**Informed Consent Statement:** Not applicable.

**Data Availability Statement:** The data presented in this study are available on request from the corresponding author.

**Conflicts of Interest:** The authors declare no conflict of interest.

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
