# Peer review of "Characterizations and Kinetic Modelling of Boride Layers on Bohler K190 Steel"

_coatings, doi:10.3390/coatings13061000_

Round 1
Reviewer 1 Report
The manuscript "Characterizations and kinetic modelling of boride layers on Bohler K190 steel" investigates the boronizing process which was carried out in the range of 1173 to 1323 K, for 10 h. The samples were boronized in solid medium, called the Durborid powder mixture. For the microstructural observations, the scanning electron microscopy has been utilized for determining the morphology of interfaces and measuring the layers’ thicknesses. This manuscript is worth publishing as it is a useful technical documents. Yet, there are several issues which need to be addressed, before it can be accepted.
1. The most important issue is the introduction and discussions on the modelling. The introduction on the fatigue of alloys and structures is rather poor. Authors have presented a straightforward and rather simple diffusion-based model. Yet, there several other more advanced modeling strategies, when it comes to the surface alloying via diffusion processes. I strongly advise authors consult and include following articles in their revisions, as this might be useful for readers who would like to go deeper in the modeling part. This following manuscript is recommended to be checked:
- Chen, L., & Zhao, Y. (2022). From classical thermodynamics to phase-field method. Progress in Materials Science, 124, 100868. doi: https://doi.org/10.1016/j.pmatsci.2021.100868
2. Dear authors, I think you need to add a paragraph about the importance and the necessity of doing such treatments in steel structures. This is a technical paper, and therefore this explanation could be very useful. Please refer to the following article, as an example:
- Jiang, J., Ye, M., Chen, L. Y., Zhu, Z. W., & Wu, M. (2023). Study on static strength of Q690 built-up K-joints under axial loads. Structures, 51, 760-775. doi: https://doi.org/10.1016/j.istruc.2023.03.034
3. When it comes to the diffusion in this manuscript, I feel that the paper needs a more in-depth discussion. For instance, the inter-diffusion and the diffusion length as well as the diffusion mechanisms are hardly discussed. The following manuscript could be very useful in adding more discussion to the manuscript.
- Yang, J., Shang, L., Sun, J., Bai, S., Wang, S., Liu, J., Yun, D., Ma, D. (2023). Restraining the Cr-Zr interdiffusion of Cr-coated Zr alloys in high temperature environment: A Cr/CrN/Cr coating approach. Corrosion Science, 214, 111015. doi: https://doi.org/10.1016/j.corsci.2023.111015
4. If possible, please provide EDS line scan from the surface down to the bulk.
5. The part related the calculation of boron diffusion coefficient was very nice. Well done.
6. Please add scale bar to micrographs in Figure 4.
7. Please provide standard chemical composition of the steel used in this investigation.
8. Is “metallographically prepared” a correct term? Please check.
9. The first sentence of your abstract is a bit strange: “In this study, the Bohler K 190 steel was used.” I suggest you start with what you have been planning to do in your research.
10. “Chromium was found to be more soluble in Fe2B than in FeB”. Why? Please explain.
Author Response
Dear reviewer, we thank you for your comments and suggestions. The replies on them are in attached file. Kind regards. Authors

Reviewer 2 Report
This paper deals with the ¨ Characterizations and kinetic modelling of boride layers on Bohler K190 steel¨. The manuscript topic is interesting, but a couple of issues can be detected in this article that caused the reading and following the manuscript to be hard:
1- The abstract is confusing. It is hard to follow the author's approach in their research. Please add the exact methodology and more results.
2- There are typos and grammatical errors.
3- The authors should state what kind of problem they want to solve. This review paper's challenges, outlooks, and novelty are not clearly described.
4- Please add a table for presenting the chemical compassion
5- The coating procedure and details are almost ignored. The topic is related to the boride layers on the steel. More information with pictures of the equipment should be added to the manuscript.
6- Add coated samples in the experimental part and a schematic view of the sections you presented in the results part. (SEM, XRD, Hardness)
7- Detailed modeling section should be detailed and described in a separate section. After that, the output can be presented in the results section. The authors give equations in results without any output. It is not clear what their aim was to present section 3.1.
8- Figures are small with low resolution. For example, the scale bar and details of the SEM image in Figure 3 are not transparent. Please replace all figures with high-resolution pictures.
9- Please add an EDS map of all samples.
10- From a general point of view, the reliability of EDS analysis for the quantification of elements is very low (uXRF or Raman spectroscopy are more robust). In this case, if the authors are interested in presenting this output, it is better to add the EDS point results in the manuscript.
11- Is Fig. 9 original, or does it need a reference? The quality is very low!!!
12- The discussion on results should be improved.
There are typos and grammatical errors in manuscript.
Author Response

(The authors gave the same response as above.)

Round 2
Reviewer 1 Report
I would like to thank authors for their efforts in their revision. The revised version is acceptable.
Reviewer 2 Report
The authors answered the comments correctly.